# Vision-Based Grasping through Goal-Conditioned Masking

## Abstract

Goal-Conditioned Reinforcement Learning for robotic reaching and grasping has enabled agents to achieve diverse objectives with a unified policy, leveraging goal conditioning such as images, vectors, and text. The existing methods, however, carry inherent limitations; for example, vector-based one-hot encodings allow only a predetermined object set. Meanwhile, goal state images in image-based goal conditioning can be hard to obtain in the real world and may limit generalization to novel objects. This paper introduces a mask-based goal conditioning method that offers object-agnostic visual cues to promote efficient feature sharing and robust generalization. The agent receives text-based goal directives and utilizes a pre-trained object detection model to generate a mask for goal conditioning and facilitate generalization to out-of-distribution objects. In addition, we show that the mask can enhance sample efficiency by augmenting sparse rewards without needing privileged information of the target location, unlike distance-based reward shaping. The effectiveness of the proposed framework is demonstrated in a simulated reach-and-grasp task. The mask-based goal conditioning consistently maintains a ∼90% success rate in grasping both in and out-of-distribution objects. Furthermore, the results show that the mask-augmented reward facilitates a learning speed and grasping success rate on par with distance-based reward.

## 1 Introduction

Acquiring a range of skills, such as reaching and grasping various objects, is one of the grand challenges for intelligent agent systems. Goal Conditioned Reinforcement Learning (GCRL) addresses this challenge by flexibly representing a range of goals, facilitating versatile skill acquisition. Unlike traditional Reinforcement Learning (RL), which targets a single task as defined by its reward function, GCRL focuses on enabling agents to simultaneously master multiple tasks using the same policy (Liu et al., 2022).

A key challenge in GCRL lies in specifying the goal condition for different tasks. Although appropriate goal conditioning can facilitate feature sharing across multiple tasks, boosting learning efficiency, the choice of goal conditioning may greatly impact the policy's ability to generalize (Kaelbling, 1993; Liu et al., 2022). Typically, the goals are defined as the desired properties or features that the agent must reach, which can be either represented as vectors (e.g., target position (Tang & Kucukelbir, 2020), orientation (Brockman et al., 2016), velocity (Zhu et al., 2021)), images (e.g., target image, Beattie et al., 2016), or text (e.g., instruction sentences, Chan et al., 2019). Using vector-based one-hot encoding as the goal conditioning provides a concise format but is constrained by the encoding's fixed size. Additionally, the encoding provides restricted information regarding the relation between goals, making it hard for the agent to build on learned skills. On the other hand, image-based goal conditioning offer the flexibility to include new objects but struggle with generalizing to out-of-distribution objects, necessitating extensive training for each new task.

The difficulty of image-based goal conditioning in generalization primarily stems from the specification being closely tied to the objects in the training sets. This specificity can hinder the agent's ability to transfer learned behaviors to new objects with different characteristics, as this goal conditioning lacks a broader, more abstract understanding of the goal itself.

Another common challenge in GCRL is that sparse reward functions, although easy to implement, impede the sample efficiency of the agent. Under such conditions, the agent must execute a long se-

quence of correct actions before receiving positive feedback (Vasan et al., 2024). Numerous studies have sought to mitigate this problem, with one popular approach being the use of a distance-to-goal metric as a dense reward function (Trott et al., 2019). However, this reward function often introduces new local optima, sometimes strongly depending on the environment and task definition, that prevent agents from learning the optimal behavior for the original task (e.g., knocking other objects along the way when reaching among multiple ones, (Trott et al., 2019; Booth et al., 2023)). Moreover, the reliance on privileged information for calculating distance-based rewards poses significant challenges in real-world applications, underscoring the need for a new approach that leverages informative dense reward signals without such requirements.

Based on the two challenges, our main contribution in this work is a novel goal conditioning based on the masking of the target object generated. The mask-based goal conditioning offers several advantages. First, It provides a *relative* goal location with respect to the current observational state, dynamically adjusting throughout the agent's interactions with the environment. Second, It enables efficient feature sharing and flexibility in adapting the trained policy to novel goal objects or locations. This is because image-based goal-conditioning by design represents a specific object, while the masking approach learns an object-agnostic way of reaching. Third, It eliminates the need for a significant amount of experience with a wide variety of goal images like that of an image-based goal conditioning. Finally, It has a lower dimension than the raw RGB image of the goal image to facilitate faster training. In a reach-and-grasp experiment, we show that mask-based goal conditioning can serve as an external augmented reward, which performs significantly better than a purely sparse reward and is comparable to a distance-based reward while eliminating the need for privileged information.

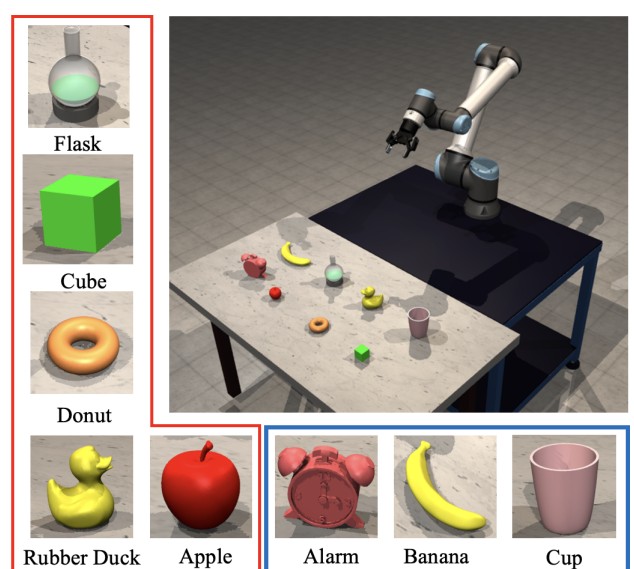

Figure 1: Simulation setup of reach and grasp, compromising a UR10e robot with 2F-85 robotiq gripper positioned on two tables, with 7 distinct objects chosen from object_sim (Dasari et al., 2023) and flask created with Blender (Community, 2018). The in-distribution training is bounded in red, while the out-of-distribution testing objects are bounded in blue.

Our contribution further includes a simulation environment. Existing simulation environments for reach-and-place tasks, such as FetchPickAndPlace from Gym Robotics (Todorov et al., 2012) and PickandPlace from Raven (Zeng et al., 2020), often employ side-mounted cameras, rely solely on vectorized proprioception inputs, or use suction grippers that restrict the types of objects that can be manipulated. These existing frameworks do not align with our GCRL setup which requires egocentric vision input and customizable observation input. Hence, we introduce a new simulation environment in MuJoCo that supports egocentric camera views, multi-input observation states, and a two-fingered parallel gripper, enhancing the versatility and realism of the simulated tasks. We plan to integrate this framework with Robohive (Rob, 2020) to serve as a benchmark for vision-based reach-and-grasp tasks[1].

---

[1]The code associated with this work will be made available upon acceptance of the paper.

## 2 RELATED WORK

GCRL has observed significant advancements in complex image-based, simulated, and real-world robotics tasks (Liu et al., 2022). Nair et al. (2018; 2020) proposed RL with Imagined Goals (RIG) that produced good reaching and pushing tasks with a Sawyer robot. Similarly, Chebotar et al. (2021) showed that through Q-learning, hindsight relabeling, and a goal-chaining mechanism, the agent learns effective goal-conditioned robotic skills. Additionally, Eysenbach et al. (2022) showed that by using Q-function and contrastive learning, RL agents can solve benchmark pushing and pick-and-place tasks. Whereas these approaches require complex modifications to policy learning, self-supervised and offline learning, ours learns from standard online RL. Moreover, each of these methods requires that the set of goal conditions be a subset of the environment states, such as the joint orientation of the desired outcome or image of the completed task. Alternatively, our method utilizes target mask goal conditions, which are much more general and easier to produce goal conditions.

More recently, Uppal et al. (2024) introduced SPIN, an RL system that can simultaneously solve navigation and pick and place tasks. The authors set the pick reward based on the distance to the target and used depth data and a pre-trained YOLO model to detect and calculate the target's position. Xiong et al. (2024) used an RL system capable of controlling a wheeled robot base and a robotic arm to reach and open doors. The authors used text-based, off-the-shelf vision models to detect doors and door handles and used sparse rewards and expert demonstrations for training. In contrast to these works, our approach is much simpler, as we only use RGB images and masks generated using pre-trained models to reach and grasp objects, and we train the RL agent from scratch. We avoided distance to target based rewards since that could lead to local optima (Trott et al., 2019; Booth et al., 2023). Hindsight Experience Replay (HER, Andrychowicz et al., 2017) is commonly used in GCRL with sparse environments. However, unlike our approach HER requires goal conditioning to be a subset of the environment state space.

Despite the success of GCRL and traditional RL systems in vision-based environments, generalizing to unknown objects and automated goal acquiring remains challenging. In this work, we address these challenges by proposing an efficient RL framework for goal-based reaching and grasping tasks.

## 3 GCRL PRELIMINARIES

GCRL augments the standard RL observation with a goal that the agent must achieve. In this work, we focus on episodic GCRL where the goal is randomly selected at the start of each episode and remains fixed until the end of the episode. GCRL is formally described by a goal-augmented Markov Decision Process (GA-MDP) (Liu et al., 2022), with the tuple $\langle \mathcal{S}, \mathcal{A}, \mathcal{T}, r, \gamma, \rho_0, \mathcal{G}, p_g, \phi \rangle$, where $\mathcal{S}, \mathcal{A}, \gamma, \rho_0$ are the state space, action space, discount factor, and distribution of the initial state. $\mathcal{T} : \mathcal{S} \times \mathcal{A} \times \mathcal{S} \rightarrow [0, 1]$ is the dynamic transition function, $\mathcal{G}$ denotes the space of goals describing the tasks, $p_g$ represents the desired goal distribution of the environment, and $\phi : \mathcal{S} \rightarrow \mathcal{G}$ is a tracable mapping function from state to goal. Here, $r : \mathcal{S} \times \mathcal{A} \times \mathcal{G} \rightarrow \mathbb{R}$ is the reward function defined with the goal. $\pi : \mathcal{S} \times \mathcal{G} \times \mathcal{A} \rightarrow [0, 1]$ is the goal-conditioned policy that maximizes the expectation of the cumulative return over the goal distribution:

$$J(\pi) = \mathbb{E}_{\substack{a_t \sim \pi(\cdot|s_t,g), g \sim p_g, \\ s_{t+1} \sim \mathcal{T}(\cdot|s_t,a_t)}} \left[ \sum_t \gamma^t r(s_t, a_t, g) \right]. \tag{1}$$

**Sparse rewards** are the most straightforward setup in real-world robotics applications. They are typically represented with a binary signal to indicate whether the task is completed:

$$r_g(s_t, a_t, g) = \mathbb{1} \text{ (Goal reached)} , \tag{2}$$

where the goal $g$ is sampled from $p_g$. In robotics settings, the goal is commonly considered satisfied when the end effector or target object is within a specific distance $\epsilon$ of the goal position:

$$\mathbb{1} \text{ (Goal reached)} = \mathbb{1}(||\phi(s_{t+1}) - g|| \leq \epsilon). \tag{3}$$

GCRL is intrinsically difficult to train under the sparse reward setting due to the lack of a meaningful signal directing the agents toward the goal object in intermediate states. This renders learning slow and exploration difficult.

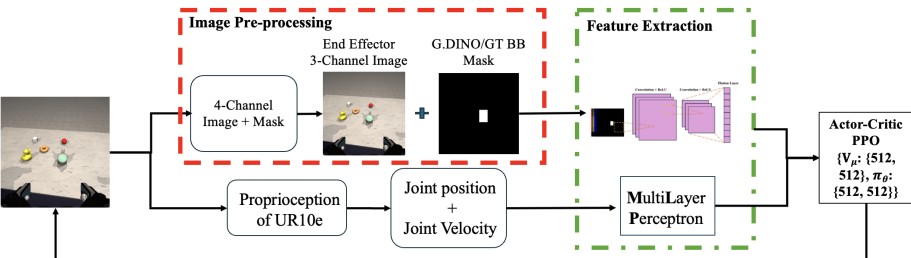

Figure 2: The framework of Goal Conditioned Reinforcement Learning with Object Mask Goal Conditioning.

To circumvent this issue, a **dense reward** function is used that is shaped according to the distance $d$ between the current location and the desired goal:

$$\tilde{r}_g(s_t, a_t, g) = -d(\phi(s_{t+1}), g).$$ (4)

## 4 PROPOSED METHOD FORMALISM

In GCRL, the agent must learn the relationship between its current observation and the episodic goal condition. The way the goal condition is specified can have a significant impact on the efficiency of learning. Goal conditions specified as a target image, for example, are high-dimensional and include a large amount of irrelevant and distracting information that leads to slow learning and poor generalization. To address this issue, we propose a goal-condition with a target object mask. The agent utilizes the text description of the target provided by the environment to generate a target object mask for goal conditioning. The goal conditioning mask is updated at each time step based on the agent's current ego-centric observation. In addition, the proposed method utilizes the size of the masked area to augment the sparse reward for improved sample efficiency. We demonstrate the proposed method based on oracle-generated masks and masks generated from the output of pre-trained grounded object detectors. For the latter, we use GroundingDINO (G.DINO, Liu et al., 2023)[2], an open-set object detector that can produce a bounding box around arbitrary objects with text descriptions such as category names.

### 4.1 MASK-BASED GOAL CONDITIONING

At the beginning of each episode, the environment selects a goal and a corresponding text string to complete the task. At each timestep, the target object text description and a copy of the current ego-centric observation are mapped to a bounding box (BB) is generated around the target object in the current frame. A one channel image is created from this where all objects outside the BB are black and those inside the BB are white[3]. The masking process is defined as

$$g_m(t) = E(o(t)),$$ (5)

where $E$ encompasses the process of (1) using a model to identify the goal object by text strings, and (2) generating a mask corresponding to the area of the bounding box. Here, $o$ is the image observation at each timestep $t$. In this work, $o$ is an egocentric view from a camera on the end effector of a robotic arm. The agent selects the next action based on the current observation, included in $s_t$, and mask: $\pi(a_{t+1}|s_t, g_m)$.

### 4.2 MASK AUGMENTED REWARD

In addition to leveraging the object mask for goal conditioning, we utilize the mask to augment the sparse reward signal. In particular, we use the change in mask size as the agent moves closer

---

[2]We selected GroundingDINO to generate the bounding box; however, other pre-trained models could also offer similar functionality.

[3]If the target object is out of view, the mask is all black.

(or farther away) from its goal, which can be seen as an approximation of standard distance-based reward functions. At each time step $t$, the **augmented reward** is calculated as

$$\hat{r}_g(s_t, a_t, g_m) = \sum_{i=1}^{M} \sum_{j=1}^{N} \phi(s_{t+1})_{ij} = \sum_{i=1}^{M} \sum_{j=1}^{N} g_m(t+1)_{ij} \, , \qquad (6)$$

where $\phi(s_{t+1}) = g_m$ is the normalized (discussed below) binary object mask generated from the BB with dimension $M \times N$. Here, the goal state produced by the tracing function $\phi$ directly aligns with that of the desired goal state $g_m$. At each time step, the agent receives a new observation from the egocentric RBG camera on the end effector and a traget mask is produced. Pixel-counting is applied to the target mask. As the agent moves closer to the goal, the masked area becomes larger and the reward increases proportionally. Thus, the pixel-counting-based reward serves as a proxy for a distance based reward shaping without the need for explicit knowledge of the objects location in the physical environment. The full reward function (sparse + augmented) is given by

$$r'_g(s_t, a_t, g_m) = r_g + \hat{r}_g = \mathcal{C}(s_{t+1}) + \sum_{i=1}^{M} \sum_{j=1}^{N} \phi(s_{t+1})_{ij} \, , \qquad (7)$$

where $\mathcal{C}(s_{t+1}) = 1$ if the goal is successfully completed at time $t + 1$, and zero otherwise. In our reaching and grasping task, this occurs when the two finger gripper grasps the target object. This masking function eliminates the need for privileged information, such as object coordinates or the distance between the achieved and desired goals, allowing for easy transfer to the real world where such information is unavailable.

The mask-based augmented reward is normalized to be in a fixed range $[0, 2]$ based on two pixel proportions. The first measures the overall proportion of white pixels within the entire $224 \times 224$ mask. The second focuses on the region of interest surrounding the gripper, aiming to ensure that the object is positioned as close as possible to the gripping position (see Fig. 3):

$$r'_g = \frac{\sum_{i=1}^{M} \sum_{j=1}^{N} \phi(s_{t+1})_{ij}}{M \times N} + \frac{\sum_{i=1}^{P} \sum_{j=1}^{Q} \phi(s_{t+1})_{ij}}{P \times Q} \, . \qquad (8)$$

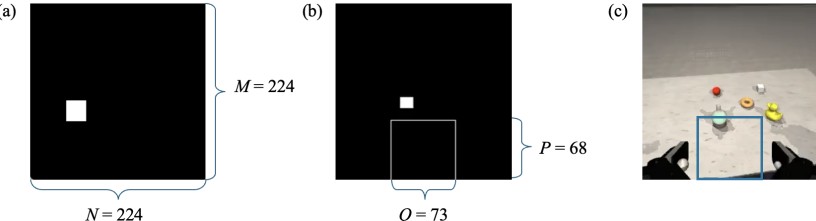

Figure 3: Calculation of mask-based augmented reward. (a) Dimension ($M \times N$) of the mask image to calculate the overall percentage of white pixels presented (b) Dimension ($P \times Q$) and location of the desired gripping location to calculate the percentage of white pixels. This is defined based on the relative position of the egocentric camera and gripper, and stays fixed throughout the experiments. (c) RGB representation of the desired gripping location.

## 5 EXPERIMENTAL SETUP

**UR10e Goal Conditioned Reaching and Grasping Environmnet:** Due to the lack of open-source environments for goal-conditioned reaching and grasping of arbitrary objects of diverse sizes and complexities, we develop a new environment to evaluate the algorithms proposed in section 4. The simulated environment includes a UR10e robotic arm with a 2F-85 gripper with 7 degrees of freedom in the MuJoCo simulator (Todorov et al., 2012). The environment has a 7D continuous action space that controls joint and gripper velocities based on the range of motion of the UR10e robot [4]. The environments has a multi-input observation space composed of the $3 \times 224 \times 224$ RGB image from the end-effector camera and UR10e 7D proprioception. The camera is positioned on top of the

---

[4]https://www.universal-robots.com/products/ur10-robot/

gripper wrist with a field of view of 65 deg$^2$. The target object for reaching and grasping is randomly selected at the start of each episode from the fixed set shown in Fig. 1, and the goal conditioning is generated as one-hot encoding, $3 \times 224 \times 224$ target image, or $1 \times 224 \times 224$ binary mask as dicussed below.

The target objects are placed on a table in front of the UR10e robotic arm and within the initial view of the end-effector camera. At the start of each episode, the positions of the five objects are randomly swapped, and they are collectively translated by a small, random distance along the $x$ and $y$ axes. The task is considered successfully completed when both pads of the gripper make contact with the goal object. The maximum length of the episode is set to 250 steps.

**Algorithms and Evaluation:** For our experiments, we train the agent using an on-policy algorithm, Proximal Policy Optimization (PPO, Schulman et al., 2017) implemented in stable-baselines3 (Raffin et al., 2021). The PPO hyper-parameters are included in Appendix A.1.

A ground truth oracle and G.DINO are applied to generate the goal conditioning mask in our experiments. They are selected to demonstrate the strength of our method with perfect masking and its implementation with a pre-trained object detector. Image resolution has a strong influence on the accuracy of G.DINO. Hence, it recieves higher resolution—$3 \times 800 \times 800$ pixels—images for inferences, whereas the RL policy is limited to $3 \times 224 \times 224$ pixels. We set the G.DINO inference threshold to 0.55 to balance the true and false positive rates.

We evaluate the proposed mask-based goal-conditioning and reward augmentation strategy in terms of the mean and standard error of the return and episode length averaged over 10 random seeds, as well as the reaching and grasping success rate on in- and out-of-distribution objects. We compare our approach against the standard goal-conditioning methods: one-hot encoding and target image, with dense and sparse reward setups. For the evaluation of the grasping success rate of the optimally seeded policy, we define successful grasping based on a criterion of single gripper contact.

## 5.1 EXPERIMENT 1: GENERALIZABILITY OF TARGET OBJECT MASKING FOR GOAL CONDITIONING

In our first experiments, we cross-compared the proposed mask-based goal condition with standard goal conditioning in terms of generalization over five in-distribution objects and evaluate on three out-of-distribution objects (see Figure 1). Here, we use a distance-based reward in order to focus solely on the goal-conditioning. As shown in Figure 4, the goal conditioning setups are:

1) **Vector-based goal-conditioning:** a one-hot encoding of the 8-element array. This provides space for the five training objects and three out-of-distribution objects.

2) **Image-based goal-conditioning:** A $3 \times 224 \times 224$ pixel generic image of the goal object that is selected at the start of each episode is appended to the observation at each timestep (Figure 1).

3) **Mask-based goal-conditioning:** A $1 \times 224 \times 224$ binary pixel mask of the target object. The experiments include two setups: *i*) ground truth (GT) target object masks generated by a bounding box (BB) oracle, and *ii*) masks generated from BB inferences produce by G.DINO using the text specification of the goal object. The ground-truth BB are generated within the MuJoCo simulation by transforming the object's coordinates into pixel points on the camera's view.

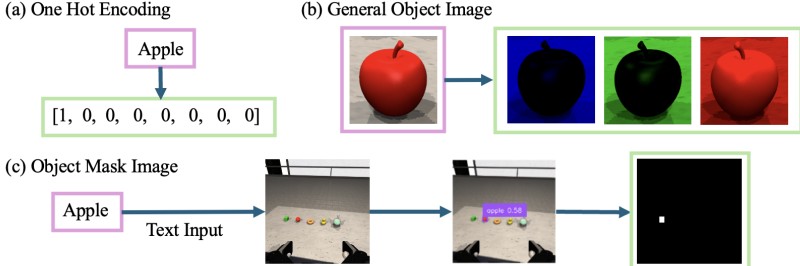

Figure 4: Three different goal conditioning for reach-and-grasp task when apple is chosen as the target object. The green bounding box shows the final goal conditioning representation.

Table 1: Comparison of goal conditioning methods and their grasping success rates[5] for in-distribution and out-of-distribution objects. The table shows the performance metrics for various methods, including Sparse and Distance-based rewards, against the Proposed Mask-Based Reward.

| Abbreviation | Goal Conditioning | Reward | Grasping Success Rate | |
|---|---|---|---|---|
| | | | In-distri | Out-distri |
| *Sparse-3C* | Object Image | Sparse | 0 | 0 |
| *Sparse-4C-GT* | GT Masking | | 0 | 0 |
| *Distance-1H* | One-Hot encoding | Distance-based Reward | 0.13 | 0.2 |
| *Distance-3C* | Object Image | | 0.62 | 0.28 |
| *Distance-4C-GT* | GT Masking | | 0.89 | 0.9 |
| *Mask-3C-GT* | Object Image | Proposed Mask-Based Reward | 0 | 0 |
| *Mask-4C-GT* | GT Masking | | 0.99 | 0.88 |

## 5.2 EXPERIMENT 2: EFFECTIVENESS OF USING MASKING IMAGE AS AUGMENTED REWARD

In our second experiment, we evaluate the reach and grasp task performance using the mask-based augmented reward $r'_g$. The augmented reward is used for training with both image-based and mask-based goal conditioning. We aim to show that the augmented reward introduced in Eq. 7 would match the performance of the distance-based reward without relying on any privileged information. The learned policy is also tested on three out-of-distribution objects.

## 6 EXPERIMENTAL RESULTS

### 6.1 GENERALIZABILITY OF GROUND TRUTH OBJECT MASKS FOR GCRL

**Goal Conditioning Results:** The results comparing the performance of the proposed method with GT masking to the traditional goal conditioning approaches (one-hot encoding and target image) with distance-based rewards are shown in Figure 6 and Table 1. Figure 6 depicts the learning curves while the grasping success rate for in- and out-of-distribution objects are shown in the Distance-based Reward section of Table 1. The standard methods of goal conditioning based on vector representation with One-Hot encoding and general image of the goal objects achieve sub-optimal returns. These goal conditionings learn to maneuver towards the goal's approximate location but fail to identify the correct target or fail to grasp it successfully with the inner pads of the gripper (Fig. 5, column 1 & 2). Specifically, when using a general

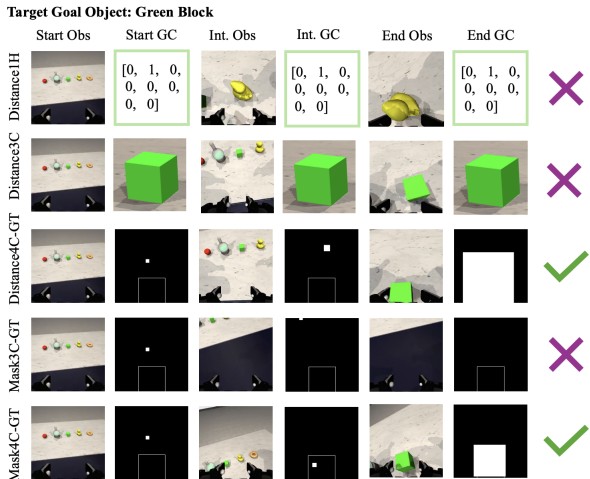

Figure 5: Visualization of the goal conditioning (GC) and observation during the start, intermediate, and ending steps of the episode. The last column indicates whether or not the reach-and-grasp is successful.

goal image for goal conditioning, although the in-distribution object grasping success rate reaches 62%, it quickly drops to 28% for out-of-distribution objects, likely due to the model's inability to learn higher-level feature abstractions that can be shared across different objects in the training set. Alternatively, our proposed mask-based goal conditioning approach increases the total return by $\sim 25\%$ and learns faster. Moreover, our approach demonstrates robustness in grasping in-distribution object, reaching 89%. Notable, the grasping performance for out-of-distribution objects remains on par with that of the in-distribution objects $\sim 90\%$.

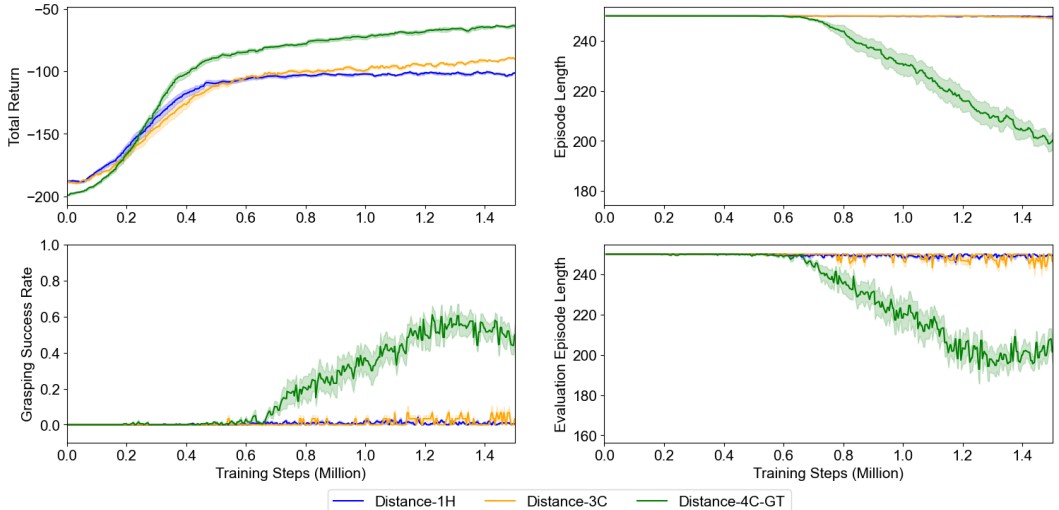

Figure 6: Comparison of different goal conditioning methods for a reach-and-grasp task using distance-based rewards, reported with standard error. The grasping success rate is additionally measured during in-training evaluation over 20 trials per session with success criteria being double contact of both grippers with the object, with the evaluation episode length representing the average episode length across those trials.

**Mask Augmented Reward Results:** This section compares the results between GT mask-based goal conditioning and standard target image-based goal conditioning using our proposed target mask augmented reward. The performance of each goal conditioning approach with the augmented reward is shown in Fig. 7 while their grasping success rate for in- and out-of-distribution objects are shown in the Dense 2 section of Table 1. When the mask-based goal conditioning is taken out of the observation state, such that the mask-based augmented reward only works in combination with the target object image goal conditioning, the agent fails to learn any behavior, even for simple reaching (Fig. 5, column 4). This is likely because, in the absence of a masking image in the observation state, the agent lacks the necessary guidance to learn to associate the target object in the dynamic RGB image with the pixel number rewarded at each timestep. Nevertheless, the mask-based augmented reward coupled with the mask-based goal conditioning is able to achieve a grasping success rate of $\sim 99\%$, even higher than the scenario when using the distance-based reward. For the out-of-distribution objects grasping rate, our method hits $\sim 88\%$, on par with the distance-based rewards.

## 6.2 TARGET MASKING WITH G.DINO FOR GCRL

Additionally, we demonstrate the use of G.DINO for mask generation in the proposed GCRL framework. This enables the agents to utilize knowledge from the pre-trained model in the observation image instead of information on the location of the target object. The results are summarized in Table 2. We analyze the grasping success rates when using G.DINO generated goal condition masks, comparing policies trained with G.DINO-derived data versus those using GT data. Our results indicate a higher grasping success rate with policies trained with GT masks compared to those trained on G.DINO generated masks in both distance-based and mask-augmented reward settings. This discrepancy is primarily attributed to the inherent noise in G.DINO inferences that can incorrectly identify the target object, leading to masks that direct the agent toward an erroneous object.

**Goal Conditioning Results:** To further elucidate the impact of noise in G.DINO inferences, we examine the agent's performance with out-of-distribution objects across three scenarios: (1) the goal object is presented alone and randomly positioned on the table; (2) one additional distractor object is introduced; (3) two additional distractor objects are introduced. We observe a monotonic decrease in the grasping success rate for distance-based reward tasks (Table 2), illustrating that the presence of additional objects introduces noise and complicates G.DINO's ability to accurately identify the

---

[5]During grasping rate evaluation, we use single-gripper contact as the success condition.

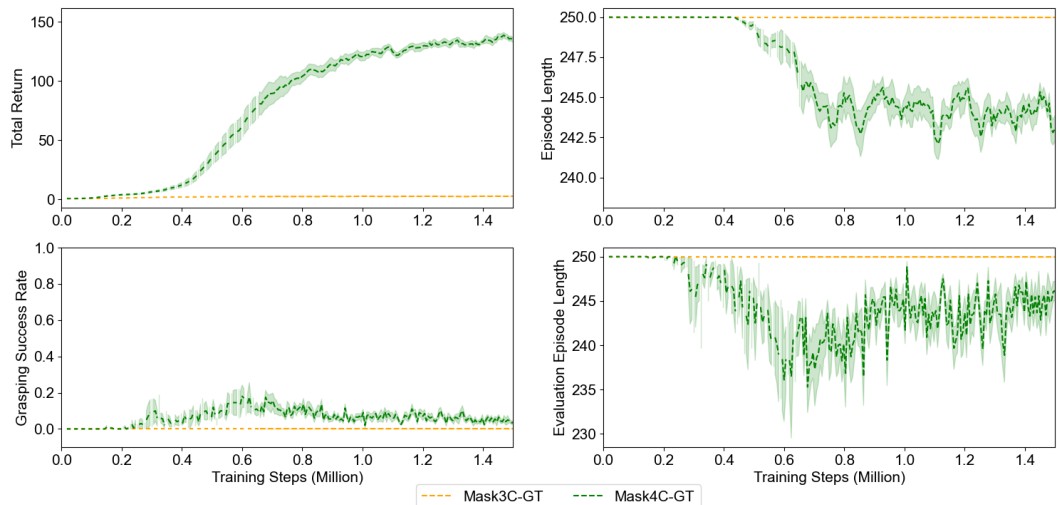

Figure 7: Comparsion between different goal conditioning methods of a reach-and-grasp task using mask-based reward tasks with standard error. The goal conditioning of each environment can be found in Table 1. All labels follows that of Fig. 6.

Table 2: Grasping success rate in an environment with G.DINO inferred mask for goal conditioning evaluated with policies trained with either G.DINO (GD) or Ground Truth (GT) Masking. For out-of-distribution objects, we evaluate the grasping success rate when 1, 2, or 3 objects are presented on the table.

| | | Grasping Success Rate | | | |
|---|---|---|---|---|---|
| **Abbreviation** | **Evaluation Policy** | In-distribution | Out-distribution | | |
| | | | 1 | 2 | 3 |
| *Distance-4C-GD* | G.DINO Masking | 0.21 | 0.28 | 0.22 | 0.24 |
| | GT Masking | 0.9 | 0.82 | 0.79 | 0.67 |
| *Mask-4C-GD* | G.DINO Masking | 0.33 | 0.22 | 0.17 | 0.08 |
| | GT Masking | 0.59 | 0.73 | 0.66 | 0.71 |

target object. Notably, when only the target object is present, the trained policy achieves a success rate of approximately 92%.

**Mask Augmented Reward Results:** For tasks using mask augmented rewards, the performance of the policy is more susceptible to false positives generated via the pre-trained model and in general performs worse than the distance-based reward environments. However, when using the policy trained with the GT goal conditioning masking, the agent can achieve an out-of-distribution object grasping success rate of $\sim 70\%$, matching that of a distance-based reward. More importantly, our results show that policies trained on GT based goal conditioning masks can be directly transferred to an environment that uses G.DINO to generate masks at test time, hence alleviating issues of extended training duration and the occurrence of false positives.

## 7 DISCUSSION AND FUTURE WORK

**Applications Beyond UR10e Reach-and-Grasp** In this work, we proposed that a pre-trained bounding box model generated by GroundingDINO can be used as an object-agnostic goal conditioning as well as an augmented reward function that is external to the environment. We presented here the application of our proposed algorithm on a UR10e robotic arm performing a reach and grasp task. However, we anticipate that this approach will also apply to other robotic arms (e.g., Franka, UR5e, Panda) and navigation robots. Moreover, this algorithm can potentially be used on a

fixed-location camera positioned close to the goal of training mobile robots. For example, teaching a quadruped robot to kick/push soccer into the net to achieve animal-level agility.

**G.DINO Limitations** The grasping success rate of using G.DINO inference is only $\leq 60\%$ of the runs using the GT masking(Table 2). The more objects that are presented on the table, the lower the grasping success rate, as the pre-trained model has a higher chance of generating false positive target masking. This is likely because G.DINO was trained from images of distant views. Hence, the accuracy of the BB model decreases accordingly when the object's completeness, angle, and distance are changed w.r.t. an ego-centric camera. Given the current limits in detection accuracies in the G.DINO pre-trained model, several directions for improvement could be considered. One potential way is to retrain the inference model on a subset of objects relevant to our task. While this may enhance performance for specific objects, it restricts the model's generalizability to objects within the fine-tuned distribution. Another strategy involves averaging the outputs of referenced objects and selecting the most frequently predicted one. However, this approach remains susceptible to noise introduced by false positives, which could impact overall reliability. Further explorations of other pre-trained models for ego-centric images might be necessary for high-accuracy inferences.

Another limitation of G.DINO is its significant time cost during inference loops. To alleviate this issue, we consider the use of asynchronous learning (Gu et al., 2016; Yuan & Mahmood, 2022) for real-time inference as part of our future work. This method has been demonstrated by Yuan & Mahmood (2022) to substantially outperform sequential learning, particularly when learning updates are computationally expensive. If the accuracy of the inferences is high, this approach could potentially expedite training and minimize the reliance on privileged information regarding object locations.

**Generalizing to Object with Versatile Shapes and Sizes** The shape and size of the target object influence the outcomes of the reach-and-grasp task. To evaluate the generalizability of our trained policies, we have intentionally incorporated objects of diverse sizes and shapes. Generally, larger objects yield broader masking inferences, which increase the rewards they receive. However, these objects can be more challenging to grasp with the gripper (e.g., banana, rubber duck). Conversely, smaller objects, though potentially more difficult for the model to infer and thus yield smaller rewards, are typically easier for the gripper to reach and handle (e.g., apple, block).

**Sim-to-Real Transfer** Our problem formulation allows easy transfer of our trained policies to real-world robotic reach-and-grasp tasks, as our mask-augmented reward does not require privileged information from the simulation. In the future, it would be interesting to extend our framework to a suite of robotic arms (e.g., Franka Arms) and further investigate sim-to-real transfer for mask-based goal conditioning with mask-based reward augmentation.

# 8 CONCLUSION

In this work, we proposed mask-based goal conditioning as an efficient representation for goal-conditioned reinforcement learning and augmented reward signals. Our method employs a pre-trained object detection model, GroundingDINO, to generate a bounding box around the goal object, which is transformed into a binary mask that feeds into the observation. We evaluated our framework in a custom-built environment using a UR10e robotic arm for a reach-and-grasp task. The results demonstrated that our proposed framework enables more efficient feature sharing across multiple goal objects and allowed robust generalization to out-of-distribution objects, outperforming traditional goal conditioning like one-hot encoding or generic object images. Furthermore, the mask-based augmented reward, which does not rely on privileged simulation information, achieves comparable performance to distance-based rewards. While current grasping success rates using GroundingDINO inference are affected by false positives in object detection, we aim to address this issue and further develop an end-to-end reach-and-grasp solution in our future work.

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

# A APPENDIX

## A.1 HYPERPARAMETER TUNING

We present here the choice of hyperparameter for different goal conditioning. The learning rate and clip range were determined after a hyperparameter sweep, ranging from [1e-4, 5e-4] and [0.0, 0.1] respectively to avoid slow convergence and catastrophic unlearning, which is a common phenomenon in PPO. We experimented with both constant and linearly scheduled decreasing rates. The entropy coefficient was set at 0.01 to balance exploration and exploitation. The neural network size is determined by the channel dimensions of the images processed during the RL training loop. Using either a one-hot encoding or a mask image for goal conditioning requires a smaller neural network to achieve a robust policy. We set the maximum episode length at 250 to allow the agent sufficient time to fully explore the table area while maintaining efficient training sessions.

Table 3: Hyperparameters of the PPO neural Network. GT = Ground Truth Masking, GD = GroundingDINO Masking, ls($x$) = linear schedule of decrease from $x$ to 0 over the entire training steps.

| Hyperparameter | Sparse-3C | Sparse-4C-GT |
|---|---|---|
| Learning Rate ($\alpha$) | ls(1e-4) | |
| Clip Range | ls(0.1) | |
| Entropy Coefficient | 0.01 | |
| Activation | ReLU | |
| Neural Network Size | 1024 | 512 |
| Max Episode Length | 250 | |

| Hyperparameter | Distance-1H | Distance-3C | Distance-4C-GT | Distace-4C-GD |
|---|---|---|---|---|
| Learning Rate ($\alpha$) | ls(3e-4) | ls(2e-4) | ls(3e-4) | ls(2e-4) |
| Clip Range | ls(0.1) | | | |
| Entropy Coefficient | 0.01 | | | |
| Activation | ReLU | | | |
| Neural Network Size | 512 | 1024 | 512 | 512 |
| Max Episode Length | 250 | | | |

| Hyperparameter | Mask-3C | Mask-4C-GT | Mask-4C-GD |
|---|---|---|---|
| Learning Rate ($\alpha$) | ls(2e-4) | | |
| Clip Range | ls(0.1) | | |
| Entropy Coefficient | 0.01 | | |
| Activation | ReLU | | |
| Neural Network Size | 1024 | 512 | 512 |
| Max Episode Length | 250 | | |

## A.2 IMAGE AND MASKING FOR GOAL CONDITIONING

We additionally run experiments on a 4-channel $224 \times 224$ pixel image combining the object image with the masking generated by Ground Truth BB or G.DINO inferened. This approach provides enriched information about the target object and captures its dynamic changes across successive timesteps. We show that although this choice of goal conditioning provides robust policies for in-distribution objects, the performance drops significantly for objects beyond the initial training set (Table 4). This might be attributed to the fact that the choice of neural network still relies heavily on the goal object image, which shows poor generalization to out-of-distribution objects.

Table 4: Comparison of goal conditioning methods and their impacts on grasping success rates for in-distribution and out-of-distribution objects. The table shows the performance metrics for various methods, including Sparse and Distance-based rewards, against the Proposed Mask-Based Reward.

| Abbreviations | Goal Conditioning | Reward | Grasping Success Rate | |
|---|---|---|---|---|
| | | | In-distri | Out-distri |
| *Distance7C-v1* | Object Image + GT Masking | Distance-Based Reward | 0.98 | 0.8 |
| *Mask7C-v1* | Object Image + GT Masking | Proposed Mask-based Reward | 0.9 | 0.27 |

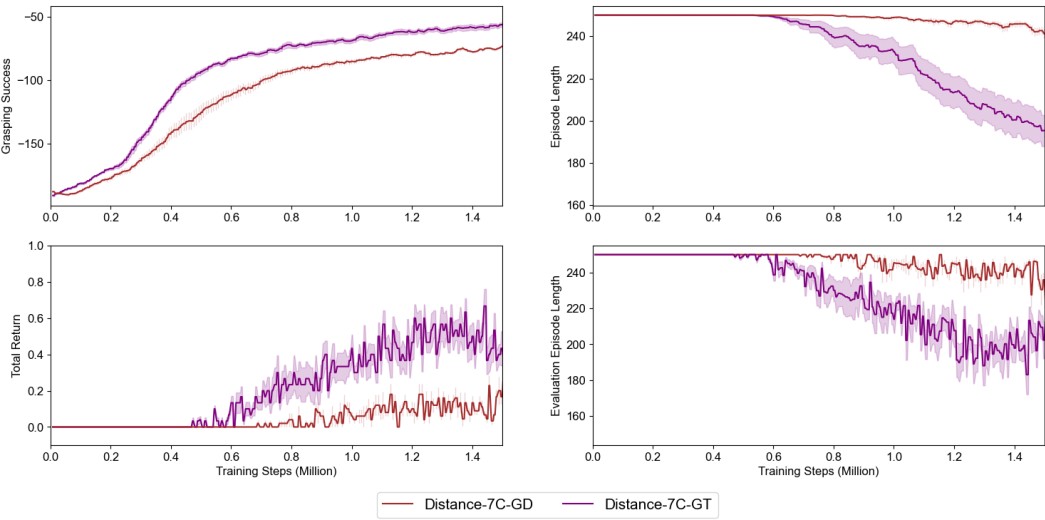

Figure 8: Comparsion between image + mask for goal conditioning using Ground Truth and G.DINO to generate BB masking in a distance-based reward. All labels follows that of Fig. 6.