# OpenReview forum: "Vision-Based Grasping through Goal-Conditioned Masking"
_ICLR.cc/2025/Conference — ICLR 2025 Conference Withdrawn Submission_

### Official Review · Reviewer_WwEv · 2024-11-01

**Soundness:** 2
**Presentation:** 2
**Contribution:** 1
**Rating:** 3
**Confidence:** 4

**Summary:**

The paper presents a novel approach to goal-conditioned reinforcement learning through the introduction of mask-based goal conditioning for robotic reaching and grasping tasks. The authors propose a method that utilizes text-based goal directives and a pre-trained object detection model to generate masks that guide the agent in achieving diverse objectives without requiring privileged information about target locations. The effectiveness of this approach is demonstrated through simulations, showing a consistent success rate in grasping both in-distribution and out-of-distribution objects. Additionally, the authors introduce a new simulation environment that accommodates egocentric vision and multi-input observation states, enhancing the realism and versatility of the evaluation framework.

**Strengths:**

The paper presents a method for goal-conditioned reinforcement learning (GCRL) that utilizes mask-based goal conditioning to enhance generalization in robotic grasping tasks. While the approach is technically sound, its originality is somewhat limited, as it builds upon existing concepts in GCRL without introducing significant new insights or paradigms. The clarity of the methodology is commendable, and the experimental results demonstrate a reasonable success rate in grasping tasks, indicating some degree of quality and significance.

**Weaknesses:**

The primary weakness of this paper lies in its lack of novelty. While the mask-based goal conditioning approach is an interesting modification, it does not fundamentally advance the field of GCRL. The work primarily repackages existing techniques without offering substantial improvements or innovative theoretical contributions. Furthermore, the experimental evaluation could benefit from a more comprehensive comparison with other established methods, as the current benchmarks do not sufficiently highlight the unique advantages of the proposed approach.

**Questions:**

1. Novelty of Contributions: Can the authors clarify how their approach significantly differs from or improves upon existing GCRL techniques? What specific gaps in the literature does this work address?

2. Baseline Comparisons: Will the authors consider including more detailed comparisons with state-of-the-art methods to demonstrate the competitive performance of their approach?

---

### Official Review · Reviewer_q2bW · 2024-11-01

**Soundness:** 2
**Presentation:** 2
**Contribution:** 2
**Rating:** 3
**Confidence:** 5

**Summary:**

The authors employ a detection model, such as Grounded-DINO, to define a dense reward function for a goal-conditioned grasping task. They have also conducted reinforcement learning experiments within their custom environment.

**Strengths:**

The authors perform comprehensive experiments, comparing various design choices and configurations.

**Weaknesses:**

1.	The paper lacks comparisons with related baselines, such as previous reward-shaping techniques.
2.	The proposed methods rely on the ground-truth bounding box of target objects, which limits feasibility for real-world applications.
3.	The methods would benefit from evaluation in a wider range of environments to demonstrate robustness.

**Questions:**

Could the target object input be specified with text, potentially integrating language models into the proposed pipeline?
Could pre-trained vision-language alignment models, such as OpenVLA, be used as the base model in this approach?

---

### Official Review · Reviewer_D6oq · 2024-11-02

**Soundness:** 2
**Presentation:** 1
**Contribution:** 2
**Rating:** 3
**Confidence:** 4

**Summary:**

This paper proposes using object masks as goal conditions and introduces two augmented rewards based on counting masked pixels to achieve faster convergence and improved generalization in goal-conditioned reinforcement learning (GCRL) tasks for robotics. The authors also set up a simulation environment with an eye-in-hand configuration in MuJoCo to evaluate the performance of the proposed method.

**Strengths:**

1. The paper proposes a novel use of object masks as an effective representation to mitigate the pattern gap caused by RGB images, improving generalizability compared to one-hot vector representations.

2. Experimental results demonstrate that the proposed mask-based reward leads to faster convergence and a higher grasp success rate.

**Weaknesses:**

1. As shown in Figure 5, the mask-based goal is time-variant compared to the fixed one-hot or RGB representations, which remain unchanged throughout the episode. This appears to be an unfair comparison, as the goal region in the image space changes during the rollout, potentially providing additional spatial information that guides the robot's movement. This dynamic nature of the mask-based goal may explain the improved performance.

2. The experiments are conducted with a limited set of objects, which may lead to statistically insufficient results. Expanding the object set could provide more robust evidence for the generalizability of the proposed method.

3. The results in Table 2 show a significant drop in performance when using inferred masks, highlighting the importance of mask precision. The authors should provide a more in-depth analysis of how different levels of noise in the masks affect the results. For example, the authors could evaluate the impact of randomly eroding or dilating the masks to simulate noise.

4. The authors should apply the proposed representation to existing works that use RGB or one-hot representations as goals to allow for a more objective comparison of methods.

5. The paper’s presentation could be clearer in several places. For instance, some parts are ambiguous—equations are labeled as (1), (2), etc., and this notation is reused later (lines 203-205), which may cause confusion. Additionally, the figures, such as Figure 2, lack sufficient detailed explanations and could benefit from more thorough descriptions.

**Questions:**

1. Why did you choose to use rectangular masks instead of precise object masks?

2. From a robotics perspective, using G.DINO to compute masks during the rollout process may be time-consuming. How long does it take to complete an episode?

---

### Official Review · Reviewer_ggr9 · 2024-11-04

**Soundness:** 1
**Presentation:** 2
**Contribution:** 1
**Rating:** 3
**Confidence:** 4

**Summary:**

The paper identifies the two challenging of specifying goals in the task of grasping object in a clutter with GCRL, namely the design of flexible and generalizable goal specification, and the difficulty in implementing a dense reward function.

The paper proposes to detect the object in the ego-centric view from textual command and use the mask to specify the goal. The reward is generated from the detection to train the policy. The design of the mask-based specification introduces several benefits, mainly contributing to the generalizability to unseen object or novel relative pose between the robot and the object. A new simulation environment in MuJoCo is developed for this task.

**Strengths:**

1. The paper identifies the current challenges in goal specification for vision-based robotic manipulation with textual commands.
2. The paper is generally easy-to-read.

**Weaknesses:**

To me, the contributions of the paper is below the bar of ICLR:

1. The task chosen, vision-based pick-up task in a clutter, has been studied by a series of works (*e.g.*, the Dex-Net series), and some of them even consider dexterous hands; see DexGraspNet 2.0 (Zhang *et al.*, 2024). A discussion to illustrate the difference from them and a comparison with them should be necessary.
2. The contribution of proposing a “new environment” is confusing. It should be easy to setup egocentric view cameras on existing benchmarks and simulations. Please clarify the technical contribution.
3. The object detection operation is also worth a deeper consideration. What if the object is out of view? Can this goal specification be generalized to other tasks like pick and place, where the information within a bounding box may be far from enough? Also, how efficient is this operation, and how fast can the policy loop run?
4. The evaluation is very naive and the results seem to indicate an unstable pipeline that is far from deployable in the real world.

Besides, a few comments:

1. The related work is not fully reviewed. Topics like GCRL, manipulation task benchmarks, and other related ones should be carefully discussed, and the related paper should be cited.
2. There should be a paragraph dedicated to describing the contributions and insights from the paper at the end of the introduction. This helps readers to rapidly identify what does this work do.

**Questions:**

1. The grasping action is produced given only 2D vision-based observations. I wonder whether this will harm the generalization, as grasping requires careful action planning based on object local geometry.
2. Could you provide a more detailed analysis of the results? Why does the evaluation based on the GroundedDINO-generated masks produce poor performance?

---

### Note · Authors · 2024-12-17

I have read and agree with the venue's withdrawal policy on behalf of myself and my co-authors.